# Adipokine-Cytokine Profile in Patients with Unstable Atherosclerotic Plaques and Abdominal Obesity

**DOI:** 10.3390/ijms24108937

**Published:** 2023-05-18

**Authors:** Evgeniia V. Garbuzova (Striukova), Victoriya S. Shramko, Elena V. Kashtanova, Yana V. Polonskaya, Ekaterina M. Stakhneva, Alexey V. Kurguzov, Ivan S. Murashov, Alexander M. Chernyavsky, Yuliya I. Ragino

**Affiliations:** 1Research Institute of Internal and Preventive Medicine–Branch of the Institute of Cytology and Genetics, Siberian Branch of Russian Academy of Sciences (IIPM–Branch of IC&G SB RAS), B. Bogatkova Str., 175/1, 630089 Novosibirsk, Russia; 2Federal State Budgetary Institution “National Medical Research Center named after Academician E.N. Meshalkin” Ministry of Health of the Russian Federation, Rechkunovskaya Str., 15, 630055 Novosibirsk, Russia

**Keywords:** abdominal obesity, unstable atherosclerotic plaque, lipocalin-2, GLP-1

## Abstract

The goal of the research was to study the levels of adipokines and their associations with unstable atherosclerotic plaques in patients with coronary atherosclerosis and abdominal obesity (AO). Methods: The study included 145 men aged 38–79 with atherosclerosis of the coronary arteries (CA) and stable angina pectoris II-III FC who were hospitalized for coronary bypass surgery (2011–2022). The final analysis included 116 patients. Notably, 70 men had stable plaques in the CA (of which 44.3% had AO), and 46 men had unstable plaques in the CA (of which 43.5% had AO). Adipocytokine levels were determined using multiplex analysis (Human Metabolic Hormone V3 panel). Results: In the subgroup of patients with unstable plaques, patients with AO had a GLP-1 level that was 1.5 times higher and a lipocalin-2 level that was 2.1 times lower, respectively. GLP-1 is direct, and lipocalin-2 is inversely associated with AO in patients with unstable plaques. Among patients with AO, the level of lipocalin-2 in patients with unstable plaques was 2.2 times lower than in patients with stable plaques in the CA. The level of lipocalin-2 was inversely associated with the presence of unstable atherosclerotic plaques in the CA. Conclusion: GLP-1 is directly associated with AO in patients with unstable atherosclerotic plaques. Lipocalin-2 is inversely associated with unstable atherosclerotic plaques in patients with AO.

## 1. Introduction

Obesity, as an independent disease or a syndrome, and related metabolic disorders are an urgent problem of modern medicine since they lead to several serious diseases development, including atherosclerosis and cardiovascular diseases (CVD) [1]. Abdominal obesity (AO) does not depend on body mass index and is determined by an increase in waist circumference (WC) of more than 94 cm for Caucasian men and more than 80 cm for Caucasian women [2]. Adipose tissue functions not only as a reservoir for fat cells but also can produce and secrete adipokines, as well as act as a receptor for various metabolically active compounds [3]. Disorders in the secretion of adipokines are the essence of the condition that is called “dysfunction of adipose tissue” [4].

Adipokines are able to support chronic inflammation [5]. Chronic inflammation, in turn, is a factor contributing to the formation of atherosclerotic plaque at all stages of its development. The inflammatory signaling system in atherosclerosis coordinates a set of macrophages and T-lymphocytes, which significantly affect the stability of the plaque, leading to the development of unstable plaque prone to rupture and thrombosis [6]. There is no doubt that obesity is a significant risk factor for the components of metabolic syndrome development [7], but his relationship is not unambiguous. Regardless of the severity of metabolic disorders in the course of obesity, obesity is associated with increased oxidative stress. It has been shown that obesity and insulin resistance are the components of metabolic syndrome that have the strongest influence on the development of oxidative stress in people with metabolic syndrome. Oxidative stress plays a significant role in the development of atherosclerosis. Recently, there has been much discussion about the importance of oxidatively modified lipoproteins (e.g., nitrated lipoproteins) in the development of cardiovascular dysfunction [8].

Many studies combined in a large review mention the relationship between glucose-dependent insulinotropic polypeptide (GIP) and glucagon-like peptide-1 (GLP-1) and inflammation, emphasizing the importance of GIP and GLP-1-dependent pathways in atherosclerosis and coronary heart disease, and say that GIP and GLP-1 can be used as markers of the frequency, clinical course and recurrence of coronary heart disease and are associated with the degree and severity of atherosclerosis and myocardial ischemia [9]. Numerous studies have shown that insulin resistance is a strong predictor of atherosclerotic CVD [10,11]. Leptin has a powerful proatherogenic effect on many types of vascular cells, including macrophages, endothelial cells, and smooth muscle cells; these effects are mediated by the interaction of leptin with the leptin receptor, which is abundantly expressed in atherosclerotic plaques [12]. Adiponectin is a multifaceted biomarker that can have a beneficial effect on the processes of atherosclerosis, inflammation, and insulin resistance [13]. Adipsin inhibits lipid uptake in a PPARy/CD36-dependent manner and prevents the formation of foam cells; it is suggested that adipsin may be a potential therapeutic target against atherosclerosis [13]. Lipocalin-2 deficiency promotes the growth of the lesion in the early stages of the disease, while it reduces the activity of MMP-9 and the size of the necrotic nucleus in progressive atherosclerosis [14]. Resistin binds to specific receptors such as receptor 4 (TLR4) or adenyl cyclase-associated protein 1 (CAP1). By binding to TLR4 and CAP 1, resistin can trigger various intracellular signaling pathways, causing vascular inflammation, lipid accumulation, and instability of atherosclerotic plaques [15]. Thus, the aim of our study was to study the levels of adipokines (Glucose-dependent insulinotropic polypeptide (GIP), Glucagon-like peptide-1 (GLP-1), Glucagon, Insulin, Leptin, Adiponectin, Adipsin, Lipocalin-2, Resistin), as well as their associations with unstable atherosclerotic plaques in patients with coronary atherosclerosis and abdominal obesity.

## 2. Results

### 2.1. Characteristics of Patients

Table 1 shows the initial characteristics of patients depending on the type of plaque (stable/unstable) and the presence of AO. Groups of men with and without AO were comparable in age, SBP, DBP, smoking status, the presence of type 2 diabetes (there were no patients receiving GLP-1 agonists), creatinine level, and GFR regardless of the type of plaque. Smokers were those who currently smoke at least one cigarette a day. All patients had established hypertension and took antihypertensive drugs until reaching the target blood pressure values. 

The presence of dyslipidemia in patients of both groups was taken into account when the level of lipids and lipoproteins increased above the optimal value [16]. Based on the very high risk of all patients included in the study, dyslipidemia was established at an LDL level > 55 mg/dL, as well as at a TG level > 150 mg/dL. Regardless of dyslipidemia in the anamnesis, all patients with coronary artery disease received high-intensity statin therapy in the maximum tolerated dosages (rosuvastatin 20–40 mg and atorvastatin 40–80 mg).

Table 2 shows the content of adipokines depending on the presence of AO in patients with stable and unstable atherosclerotic plaques.

### 2.2. Differences between Subgroups of Patients with AO and without AO in Patients with Unstable Plaques

Table 2 also shows the content of adipokines depending on the presence and absence of AO in patients with unstable atherosclerotic plaques. When analyzing subgroups of patients depending on the type of plaque and AO, no differences were obtained between patients with stable plaques with AO and without AO. In the subgroup of patients with unstable plaques, patients with AO had a GLP-1 level that was 1.5 times higher (502.78 [355.71; 945.32] vs. 17.04 [13.01; 33.24], *p* = 0.044) and a lipocalin-2 level that was 2.1 (208.24 [157.64; 430.48] vs. 440.72 [205.98; 790.07], *p* = 0.027) times lower, respectively.

At the next stage, all the studied parameters were included in the model of logistic regression analysis of AO and adipocytokine associations in patients with unstable plaques (Table 3). AO was taken as a dependent variable.

According to the results of logistic regression analysis, GLP-1 is directly and lipocalin-2 is inversely associated with the presence of AO in patients with unstable atherosclerotic plaques. All the studied indicators were included in the ROC analysis with an assessment of sensitivity and specificity.

Given the poor quality of the model for lipocalin-2 (AUC 0.306), the results of the ROC analysis can be concluded only for GLP-1 (Figure 1). The level of GLP-1 ≥ 396.114 pg/mL with a sensitivity of 75% and specificity of 68% is associated with the presence of AO in patients with unstable plaques in the CA (AUC 0.676, *p* = 0.044).

### 2.3. Differences between Subgroups of Patients with Stable and Unstable Plaques in Patients with AO

Figure 2 shows the content of lipocalin-2 in the blood of patients on the background of AO, depending on the type of plaque.

In patients with AO, patients with unstable plaques had a level of lipocalin-2 2.2 times lower (*p* = 0.008) than patients with stable plaques in CA.

The next stage was a logistic regression analysis of the chance of unstable atherosclerotic plaque presence on the background of AO (unstable atherosclerotic plaque presence was taken as a dependent variable), depending on the level of lipocalin-2, in which the level of lipocalin-2 was inversely associated with the presence of unstable atherosclerotic plaques in the coronary arteries (Table 4).

During the ROC analysis to determine the threshold value of lipocalin-2 for the presence of atherosclerotic plaque instability, an unsatisfactory quality model (AUC 0.306) was obtained.

## 3. Discussion

Glucagon-like peptide-1 (GLP-1) is the main incretin hormone that is secreted from the L-cells of the small intestine and is derived from the transcription product of the proglucagon gene [17]. Glucagon is known to reduce cholesterol synthesis by inhibiting HMG-CoA reductase and increasing cholesterol excretion in the liver [18]. Glucagon increases cholesterol excretion in bile by increasing the expression of ABCA1, a regulator of cholesterol excretion with bile. A decrease in hepatic expression of sterol-binding regulatory protein 1c (SREBP-1C) was observed after treatment with GLP-1 and glucagon coagonist. The coagonist of GLP-1 and glucagon also increased the expression of LDL receptors, increased the absorption of LDL from the bloodstream, and decreased the level of circulating lipids [19]. The coagonist of GLP-1 and glucagon receptors also increase energy consumption [20]. Fundamental and clinical studies of biomarkers have established the contribution of inflammation to the development of atherosclerosis [6] and that anti-inflammatory drugs (including GLP-1 receptor agonists) can improve cardiovascular outcomes [21]. Consequently, the GLP-1 agonists studied in these studies affect mechanisms that are considered central to atherosclerosis and its clinical complications.

Fasting and post-meal GLP-1 levels were studied in adults and children with diabetes, obesity, and cardiometabolic risk factors. Previously, some cross-sectional studies reported that fasting GLP-1 levels were higher in adults with DM2 and obesity, while others found no difference. A recent study reported a significant increase in fasting GLP-1 levels in obese adult women [22,23]. It is known that GLP-1 is secreted from the a-cells of the pancreas and intestinal L-cells and that the inflammatory cytokine interleukin-6 (IL-6) stimulates the secretion of GLP-1 from the a-cells of the pancreas. Since serum IL-6 levels are elevated in obese individuals, IL-6-induced GLP-1 secretion may play a role in enhancing the proliferation of β-cells to expand the reservoir of β-cells as compensation for increased insulin requirements due to the accumulation of visceral fat and subsequent exacerbation of insulin resistance [24].

Our results can also be explained by the phenomenon of GLP-1 resistance, which is caused by excessive visceral fat [25]. Grasset et al. [26] reported that gut microbiota dysbiosis caused GLP-1 resistance in obese and diabetic mice. Although the underlying mechanism by which gut microbiota dysbiosis induces GLP-1 resistance in the enteric nervous system remains unclear, this study suggests that the gut–brain axis plays a significant role in GLP-1-activated insulin secretion and gastric emptying.

Data from the study by Albinsson-Stenholm E. et al. contradict the lack of GLP-1 release as a mediator of increased glucose levels in early manifestations of insulin resistance in white people. They found that fasting levels of GLP-1 correlated positively with serum insulin in the total cohort. However, this relationship was lost when data were corrected for age, sex, fasting glucose, and BMI. This confirms the complicated relationship between BMI and insulin when age and sex are added to the equations. They speculate that increased GLP-1 levels in subjects with hyperinsulinemia can be a compensatory mechanism in insulin resistance that occurs before glucose levels are highly elevated [27].

In our study, in a subgroup of patients with unstable plaques, patients with AO had a GLP-1 level that was 1.5 times higher; GLP-1 is directly associated with the presence of AO in patients with unstable atherosclerotic plaques, while the level of GLP-1 ≥ 396,114 pg/mL with a sensitivity of 75% and a specificity of 68% is associated with the presence of AO in patients with unstable plaques in the CA. We understand that GLP-1 has well-known anti-atherosclerotic properties and that treatment with GLP-1 agonists reduces food intake, and thus these drugs are used in the treatment of obesity. However, in our sample of patients with coronary atherosclerosis, we received contradictory results, which may supplement the known information on this topic in white people. More in-depth research in this area is needed; it is possible to study the intestinal microbiota, and the genetic predisposition of GLP-1 resistance in patients with established coronary artery disease, especially those with unstable atherosclerotic plaques.

Lipocalin-2 is a small lipid-binding protein with still unknown ligands [14], which is more expressed by mature adipocytes. Despite the well-studied role of lipocalin-2 in the immune response to bacterial infections, there is also evidence of its role as a pro-inflammatory adipokine in obesity and related metabolic diseases. There is an increase in circulating lipocalin-2 in patients with atherosclerosis and coronary artery disease and coronary CVD [28]. However, despite the fact that lipocalin-2 is elevated in the atherosclerotic focus, a more detailed study by Amersfoort and colleagues revealed that mice with lipocalin-2 deficiency had larger atherosclerotic lesions suggesting that lipocalin-2 deficiency may contribute to the growth of lesions in the early stages of the disease [14]. In a recent study on patients with coronary atherosclerosis, against the background of both normal and overweight patients, we obtained data that significantly higher levels of lipocalin-2 were found in patients with coronary atherosclerosis compared to patients with normal coronary arteries. Nevertheless, there was no difference between patients with stable and unstable atherosclerotic plaques [29].

In this study, data were obtained that the levels of lipocalin-2 are 2.1 times lower in the subgroup of patients with unstable plaques and AO compared to patients with unstable plaques without AO. Lipocalin-2 is inversely associated with the presence of AO in patients with unstable atherosclerotic plaques. Among patients with AO, the level of lipocalin-2 in patients with unstable plaques was 2.2 times lower than in patients with stable plaques in the CA. If we look at the average values of lipocalin-2 in healthy people and people with coronary atherosclerosis, it can be noted that in our study the level of lipocalin-2 was several times higher, which is consistent with the generally accepted data on its pro-inflammatory properties. Thus, we obtained differences only in a subgroup of obese patients between stable and unstable plaques, which requires further study [28]. This discovery was rather illogical since it is usually believed that lipocalin-2 is secreted in inflammatory conditions and contributes to their development [30]. Interestingly, there are actually some studies describing the protective properties of lipocalin-2 in several inflammatory diseases [31,32]. The decrease in lipocalin-2 may have increased Leukotriene B4′s ability to activate monocytes, but future research should shed more light on the exact mechanisms involved in lipocalin-2-mediated differentiation and migration of monocytes [33]. Additionally, all patients in our study received high-intensity statin therapy in the maximum tolerated dosages, which can influence lipocalin-2 levels [34]. In addition, although this study did not include patients with GFR less than 45 mL/min/1.73 m^2^, lipocalin-2 may also depend on fluctuations in creatinine and renal function, which requires a more thorough analysis. The study limitations also include a small sample size and the inclusion of only males.

Thus, this study showed that GLP-1 is directly associated with the presence of AO in patients with unstable atherosclerotic plaques; the levels of lipocalin-2 are inversely associated with the presence of AO in patients with unstable atherosclerotic plaques. These results are contradictory but may bring out some issues that deserve further investigation in future research.

## 4. Materials and Methods

The design of the study is a single-stage observational study. The study was conducted within the framework of joint scientific research of IIPM–Branch of IC&G SB RAS and FSBI “National Medical Research Center named after E. Meshalkin” of the Ministry of Health of the Russian Federation. The study was approved by the Local Ethics Committees of both institutions (Protocol No. 2, dated 5 July 2011). The data and samples were collected after receiving written informed consent from all participants.

The study included 145 men aged 38–79 years (mean age 62.28 ± 8.19) with coronary angiographically verified atherosclerosis of the coronary arteries, without acute coronary syndrome (ACS), with stable angina pectoris II-III FC, hospitalized in the clinic of the FSBI National Medical Research Center named after E. Meshalkin of the Ministry of Health of the Russian Federation for coronary bypass surgery (in the period from 2011 to 2022) (Figure 3).

The inclusion criteria at the stage of preliminary selection of patients are as follows: male, diagnosis of coronary heart disease (CHD) verified by coronary angiography data, the presence of a history of myocardial infarction (MI) or episodes of stable angina pectoris documented by a description of the clinical picture of the disease, the results of ECG, and biochemical blood tests.

The non-inclusion criteria at the pre-selection stage of patients are as follows: female, the presence of ACS less than six months before admission (unstable angina or MI), clinically significant severe concomitant pathology in exacerbation (chronic infectious and inflammatory diseases, renal failure (GFR ≤ 45 mL/min/1.73 m^2^ (CKD-EPI)), respiratory failure, and liver failure), known active oncological diseases, and toxic lesion heavy metals.

During the CABG operation, strictly according to intraoperative indications, an endarterectomy was performed on the coronary artery. Further studies of histological material were carried out in the pathomorphological laboratory of the FSBI “NMRC named after E.N. Meshalkin of the Ministry of Health of Russia”. Each endarterectomy material containing intima-media of the coronary arteries was longitudinally and transversely symmetrically divided into 3–5 fragments for histological and biochemical studies. Histological analysis of fragments of intima-media of coronary arteries after the macroscopic description of samples (prevalence of atherosclerotic plaque, degree of narrowing of the artery lumen, hemorrhages in the structures of atherosclerotic plaque, calcification sites, blood clots) and standard hematoxylin-eosin and Van Gieson staining were studied on an Axiostar Plus binocular microscope. The study of fragments of intimate media revealed the presence of stable and unstable atherosclerotic plaques. The unstable plaque was differentiated according to the following criteria: the thickness of the fibrous covering is less than 65 microns, infiltration by macrophages and T-lymphocytes (more than 25 cells in the field of vision 0.3 mm), and a large lipid nucleus (more than 40%) [35].

After the intervention, six patients dropped out of the study for the following reasons: voluntary refusal to participate, inability to contact the patient, and development of complications. In 23 patients, the type of plaques could not be determined using histological analysis. The final analysis included 116 patients. Of the studied group, 70 men had stable plaques in the coronary arteries (60.3%), and 46 men had unstable plaques in the coronary arteries (39.7%).

Before the operation, biological material (blood) was taken. Biochemical studies were carried out in the Laboratory of Clinical Biochemical and Hormonal Studies of Therapeutic Diseases of IIPM–Branch of IC&G SB RAS. By multiplex analysis using the Human Metabolic Hormone V3 panel (MILLIPLEX, Darmstadt, Germany), adipocytokine levels were determined on a Luminex MAGPIX flow fluorimeter: Glucose-dependent insulinotropic polypeptide (GIP), Glucagon-like peptide-1 (GLP-1), Glucagon, Insulin, Leptin, Adiponectin, Adipsin, Lipocalin-2, and Resistin.

The study took into account demographic characteristics, anamnesis of the disease, and the presence of chronic diseases (type 2 diabetes, hypertension, myocardial infarction, acute cerebrovascular accident). Patients underwent anthropometry, including height, body weight, waist, and hip circumference. Their body mass index was determined using the following formula: BMI = Body weight (kg)/Height (m^2^); Waist circumference (WC) was measured in a standing position, midway between the lower edge of the chest and the crest of the ilium along the middle axillary line. Abdominal obesity (AO) was established at a WC of more than 94 cm for the Caucasian race [2]. In the group of patients with unstable plaques, there were 20 patients with AO (43.5%), in the group of patients with stable plaques, there were 31 patients with AO (44.3%).

The obtained results were statistically processed using the SPSS 13.0 software package. To assess the normality of distribution, we used the Kolmogorov–Smirnov test. Variables with parametric distribution are represented as arithmetic mean (M) and standard deviation (SD), with nonparametric distribution as the median, 25, and 75 quartiles. A comparative study of variables with parametric distribution was carried out using the Student’s t-test. In the case of a nonparametric distribution, the Mann–Whitney U-test (for two independent groups) and the Kruskall–Wallis criterion were used. Spearman’s rank correlation coefficient (rs) was used to analyze the dependence of quantitative features of sample data from aggregates. In the case of nominal and ordinal data scales, cross-data tables and Pearson’s χ2 criterion were used and adjusted for probability (for calculating ORs). Associations were evaluated using multiple logistic regression analysis performed under the following conditions: the dependent variable is dichotomous; independence of observations; absence of multicollinearity, i.e., situations wherein independent variables strongly correlate with each other (r > 0.9); linear dependence between each independent variable and the logarithm of the odds ratio (logarithmic coefficients); and independence of residuals. The results of multiple logistic regression analysis were presented as OR and 95% CI for OR. To assess the threshold value of the diagnostic test, ROC analysis, sensitivity, and specificity assessment were carried out. The significance level was set at *p* < 0.05.

## 5. Conclusions

GLP-1 is directly associated with AO in patients with unstable atherosclerotic plaques. Lipocalin-2 is inversely associated with the presence of unstable atherosclerotic plaques in patients with AO. Further studies are needed to study lipocalin-2 and GLP-1 in patients with coronary atherosclerosis in this patient population.

## Figures and Tables

**Figure 1 ijms-24-08937-f001:**
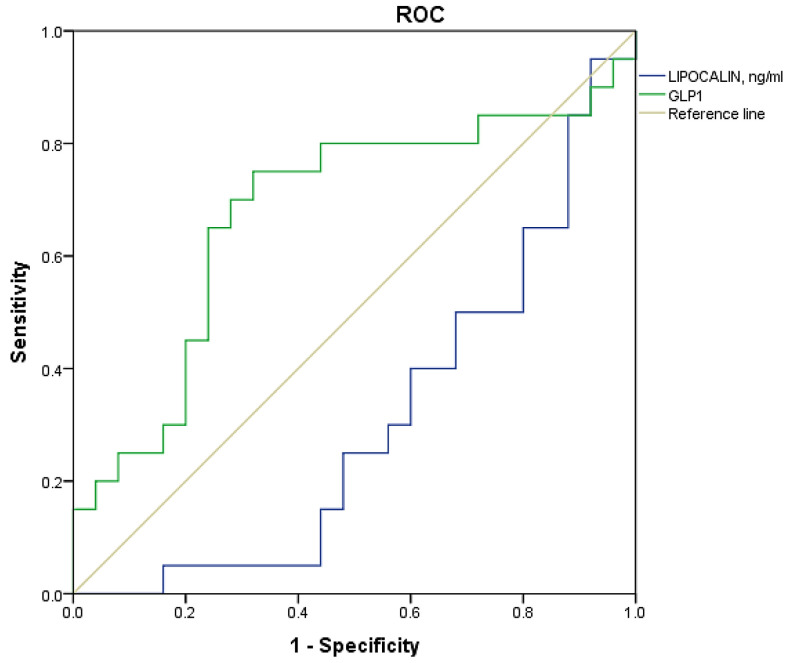
ROC curve of identification of threshold values of adipokines for AO in patients with unstable atherosclerotic plaques.

**Figure 2 ijms-24-08937-f002:**
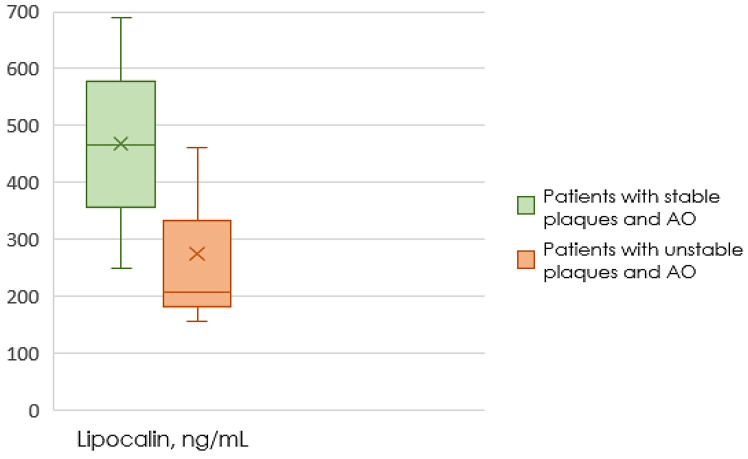
The content of lipocalin-2 in patients with AO, depending on the type of plaque.

**Figure 3 ijms-24-08937-f003:**
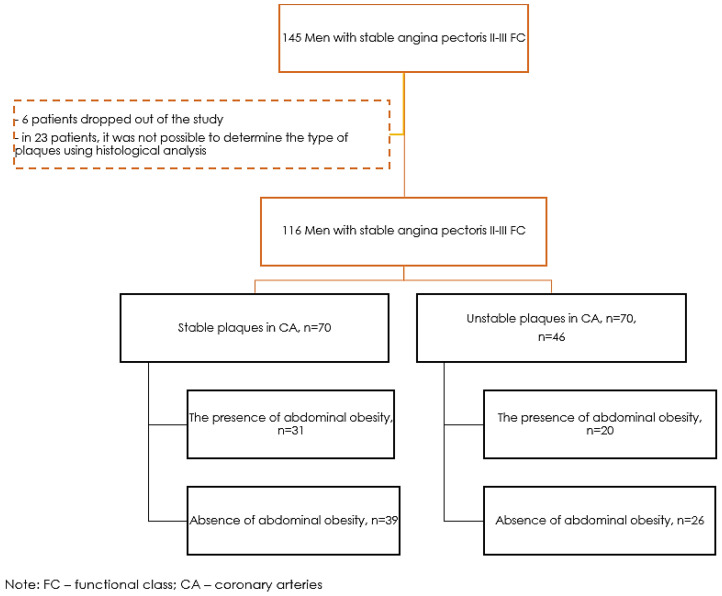
Research design.

**Table 1 ijms-24-08937-t001:** Characteristics of patient groups depending on the type of plaque (stable/unstable) and the presence of abdominal obesity.

Parameter	Patients with Stable Plaques in the Coronary Arteries		*p*	Patients with Unstable Plaques in the Coronary Arteries		*p*
	Presence of AO *n* = 31	Absence of AO *n* = 39		Presence of AO *n* = 20	Absence of AO *n* = 26	
Average age (M ± SD)	63.35 ± 8.36	61.82 ± 7.74	0.429	61.65 ± 7.78	60.92 ± 7.25	0.745
BMI, kg/m^2^ (M ± SD)	30.64 ± 4.65	26.83 ± 3.29	<0.001	31.67 ± 3.50	27.26 ± 3.56	<0.001
WC, cm (M ± SD)	99.00 ± 7.79	86.93 ± 6.09	<0.001	100.78 ± 8.32	86.38 ± 6.81	<0.001
SBP, mmHg (M ± SD)	129.89 ± 12.61	134.79 ± 12.87	0.080	135.12 ± 10.67	135.31 ± 15.59	0.963
DBP mmHg (M ± SD)	80.59 ± 9.09	82.14 ± 8.02	0.456	82.58 ± 7.62	82.50 ± 9.29	0.974
Creatinine level, µmol/L	98.18 ± 18.33	99.20 ± 17.19	0.539	99.5 ± 6.36	99.35 ± 17.03	0.923
GFR mL/min/1.73 m^2^ (CKD-EPI)	68.00 ± 10.40	72.33 ± 15.99	0.872	71.91 ± 19.88	71.60 ± 14.88	0.791
Smoking status (absolute in %)	81.5%	73.0%	0.554	85.0%	80.8%	0.999
T2DM (absolute in %)	29.6%	8.1%	0.042	30.0%	26.9%	0.999

Note: BMI-Body Mass Index, WC-Waist circumference, SBP–systolic blood pressure, DBP–diastolic blood pressure, GFR–glomerular filtration rate (calculated by the CKD-EPI formula), and T2DM-Type 2 diabetes.

**Table 2 ijms-24-08937-t002:** Adipocytokine content depending on abdominal obesity and type of atherosclerotic plaque, Me (25–75%).

Parameter	Patients with Stable Plaques with AO *n* = 31	Patients with Unstable Plaques with AO *n* = 20	*p*	Patients with Stable Plaques without AO *n* = 39	Patients with Unstable Plaques without AO *n* = 26	*p*
Glucose-dependent insulinotropic polypeptide (GIP) (pg/mL)	27.19 [19.19; 42.88]	33.39 [12.43; 43.45]	0.912	26.82 [16.31; 51.83]	17.04 [13.01; 33.24]	0.053
Glucagon-like peptide-1 (GLP-1) (pg/mL)	398.75 [212.52; 637.31]	502.78 [355.71; 945.32]	0.203	292.98 [191.27; 619.17]	334.53 [173.64; 457.93]	0.999
Glucagon (pg/mL)	10.45 [4.41; 25.35]	5.87 [3.65; 14.98]	0.130	10.07 [4.56; 20.41]	8.82 [3.74; 23.03]	0.747
Insulin (pg/mL)	340.82 [272.95; 552.49]	330.84 [272.95; 473.75]	0.894	429.78 [277.61; 591.67]	340.82 [205.02; 643.31]	0.269
Leptin (pg/mL)	5714.34 [3156.40; 8509.36]	4184.82 [1606.94; 11,380.80]	0.516	4274.68 [1118.54; 7734.57]	4354.34 [935.56; 9777.80]	0.680
Adiponectin (mcg/mL)	27.41 [9.40; 12.87]	16.76 [12.71; 34.74]	0.475	25.47 [17.34; 31.15]	28.70 [14.23; 41.06]	0.831
Adipsin (mcg/mL)	9.40 [7.37; 12.87]	10.21 [7.61; 12.31]	0.602	9.65 [6.00; 15.01]	11.34 [7.84; 14.62]	0.639
Lipocalin-2 (ng/mL)	467.21 [248.67; 689.59]	208.24 [157.64; 430.48]	0.008	374.40 [189.82; 716.05]	440.72 [205.98; 790.07]	0.603
Resistin (ng/mL)	24.08 [8.67; 44.58]	18.87 [4.86; 32.17]	0.516	40.28 [16.47; 50.76]	35.84 [13.14; 77.81]	0.749

**Table 3 ijms-24-08937-t003:** Logistic regression analysis of AO and adipocytokine associations in patients with unstable plaques.

Parameter	Model 1	Model 2 (Taking into Account Gender and Age)
Glucagon-like peptide-1 (GLP-1) per 10 pg/mL	1.010 (1.011–1.030), *p* = 0.058	1.020 (1.010–1.030), *p* = 0.050
Lipocalin-2 per 10 ng/mL	0.970 (0.951–0.990), *p* = 0.038	0.970 (0.957–0.990), *p* = 0.037

**Table 4 ijms-24-08937-t004:** Logistic regression analysis of associations of unstable atherosclerotic plaque in patients with AO.

Parameter	Model 1	Model 2 (Taking into Account Gender and Age)
Lipocalin-2 per 10 ng/mL	0.970 (0.942–0.990), *p* = 0.018	0.961 (0.942–0.990), *p* = 0.019

## Data Availability

Not applicable.

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
