# Peer review of "Adipokine-Cytokine Profile in Patients with Unstable Atherosclerotic Plaques and Abdominal Obesity"

_ijms, 2023, doi:10.3390/ijms24108937_

Round 1

Reviewer 1 Report

This is a cross-sectional study aiming to explore the association between coronary plaque stability and adipocytokines levels in subjects with and without abdominal obesity. Although the idea of this research is interesting, and the manuscript could add new information to this field, I have found the study design and the statistics wrong. In detail, the statistical models used (not cited in the corresponding section) are poorly explained and unclear (e.g., what is the dependent variable? The unstable plaque or the abdominal obesity?). Many variables that could be confounding factors have not been considered: many participants had diabetes, but their pharmacological therapy, that could influence the results (e.g., GLP-1 Analogues) is not described. Another variable that should be considered is creatinine: lipocalin- 2 (also known as NGAL) levels are strictly correlated with kidney function. Finally, I didn't find enough explanations to the results: GLP-1 has well-known anti-atherosclerotic properties, while lipocalin-2 is associated with cardiovascular disease and myocardial infarction. Lipocalin increases the matrix metalloproteinase pool, which could increase the risk of unstable plaque, not vice versa.

Author Response

Thank you for your review, it will definitely help to improve our work.

We have expanded the section "Statistical Analysis" in order to make it more clear about the statistical methods used. Also, additions were made to the text of the article indicating the dependent variable.

Patients receiving GLP-1 agonists were not included in this study.

We also entered data on creatinine and GFR levels for all groups of patients, no differences were obtained according to these indicators, and therefore they were not included in the logistic regression analysis.

We understand that GLP-1 has well-known anti-atherosclerotic properties and that treatment with GLP-1 agonists reduces food intake, and thus these drugs are used in the treatment of obesity in people with and without CVD. However, in our sample of patients with coronary atherosclerosis, we received contradictory results, which may supplement the known information on this topic in white people. More in-depth research in this area is needed, it is possible to study the intestinal microbiota, and the genetic predisposition of GLP-1 resistance in patients with established coronary artery disease, especially those with unstable atherosclerotic plaques. We have significantly expanded the discussion and also found studies with similar contradictory results, which does not allow us to make conclusions yet, but these data may be of interest for further research.

We have once again rechecked the results about lipocalin-2 and expanded the discussion, and although the data contradict well-known studies, there are also a number of studies with similar contradictory results, which does not exclude the interference of any other factors. If we look at the average values of lipocalin-2 in healthy people and people with coronary atherosclerosis, it can be noted that in our study the level of lipocalin-2 was several times higher, which is consistent with the generally accepted data on its pro-inflammatory properties. Thus, we obtained differences only in a subgroup of obese patients between stable and unstable plaques, which requires further study.

We understand that you have experienced difficulties due to poor translation quality. After editing, the article was checked by a professional translator.

Reviewer 2 Report

I have received to review the original research article entitled “Adipokine-cytokine profile in patients with unstable atherosclerotic plaques and abdominal obesity”, prepared by Evgeniia V. Garbuzova et al. submitted to the International Journal of Molecular Sciences (IF=6.208). Obesity and the metabolic syndrome and its components are a very important challenge for public health worldwide. They contribute to the development of type 2 diabetes, which is one of the most important risk factors for the development of cardiovascular diseases. Cardiovascular diseases are the leading cause of morbidity and mortality worldwide. The effort of the Authors should be appreciated for undertaking research on such an important topic. The paper is generally well prepared and in my opinion it should be considered for publication but I would like to suggest some recommendations which can further improve the quality and attractiveness of the manuscript.

1)     It is worth emphasizing that the waist circumference standards quoted in the introduction apply to the Caucasian race. (line 35)

2)     Although the term "adipocytokines" is correct, "adipokines" is currently the preferred term in the scientific literature worldwide.

3)     When writing about the endocrine function of adipose tissue, it is worth mentioning that disorders in the secretion of adipokines is the essence of the condition that is called "dysfunction of adipose tissue" in the literature. (line 36-38)

4)     I think the introduction should be extended. The Authors satisfactorily introduced the reader to the issues related to the importance of adipokines in the development of cardiovascular disease, but in my opinion, the information on obesity should be expanded. It is worth noting that although there is no doubt that obesity is a significant risk factor for the development of components of the metabolic syndrome, this relationship is not fully unambiguous. Because in the population of obese people, both metabolically unhealthy and metabolically healthy people can be distinguished. However, regardless of the severity of metabolic disorders in the course of obesity, obesity is associated with increased oxidative stress. It has been shown that obesity and insulin resistance are the components of the metabolic syndrome that have the strongest influence on the development of oxidative stress in people with the metabolic syndrome. Oxidative stress plays an important role in the development of atherosclerosis. Recently, there has been much discussion about the importance of oxidatively modified lipoproteins (e.g. nitrated lipoproteins) in the development of cardiovascular dysfunction. (10.1155/2021/9987352; 10.3390/antiox11010079)

5)     In table 1 there is the entry "smoking status". Explain what exactly that means. It is worth presenting in the table precisely how many people currently smoke, how many people used to smoke but do not currently smoke, and how many people have never smoked.

6)     It seems that Table 3 does not introduce any new information compared to Table 2, but only duplicates part of the information. Then please delete table 3 so as not to duplicate the same information.

7)     The discussion needs to be significantly broadened, because it is currently very poor. The authors mainly present basic information on the importance of selected adipokines in human biochemistry and pathobiochemistry. The results obtained in this work should be confronted with the results of other researchers in available scientific publications much more carefully and disseminated. What are the novel aspects of this work in relation to other publications should be discussed. The limitations of the study conducted should also be thoroughly discussed.

8)     The methodology of the conducted research has been precisely described. I have no additional comments.

9)     The conclusions, although correctly presented, are quite laconic. I suggest adding a few more sentences, e.g. in which the Authors will propose the most important directions for further research, which in their opinion imply the results obtained.

10) The work contains linguistic and editorial errors that should be corrected. I recommend that the text be subjected to additional linguistic proofreading before re-submitting the text for review.

11) I think the number of cited articles should be increased.

12) The list of references must be prepared in accordance with the requirements of the MDPI publishing house.

Author Response

Thank you for your thorough review, it has made our article better.

  1. We’ve added this information.
  2. We’ve changed the term "adipocytokines" to "adipokines."
  3. We’ve added this information.
  4. We agree that in the population of obese people, both metabolically unhealthy and metabolically healthy people can be distinguished [Smith GI, Mittendorfer B, Klein S. Metabolically healthy obesity: facts and fantasies. J Clin Invest. 2019;129(10):3978-3989. doi:10.1172/JCI129186]. But according to some studies, almost one-half of people with MHO developed Metabolic syndrome and had increased odds of CVD [Mongraw-Chaffin M, Foster MC, Anderson CAM, et al. Metabolically Healthy Obesity, Transition to Metabolic Syndrome, and Cardiovascular Risk. J Am Coll Cardiol. 2018;71(17):1857-1865. doi:10.1016/j.jacc.2018.02.055/]

We’ve expanded the introduction according to your comments. However, it seems to us that the information about metabolically healthy and unhealthy types of obesity does not quite fit into the concept of the article since all patients included in the study already initially have established coronary artery disease, which does not allow us to single out patients with metabolic healthy obesity. At the same time, there were patients whose BMI did not exceed normal values, while they were diagnosed with abdominal obesity by the value of waist circumference.

  1. We’ve added this information to the text.
  2. We agree with the remark and have deleted table 3 to avoid duplication of information.
  3. We have once again rechecked the results and expanded the discussion, and although the data contradict well-known studies, there are also several studies with similar contradictory results, which does not exclude the interference of any other factors.
  4. Thank you for your comment.
  5. We’ve expanded the conclusion section following your comment
  6. We understand that you have experienced difficulties due to poor translation quality. After editing, the article was checked by a professional translator
  7. We’ve increased the number of cited articles.
  8. We’ve prepared a list of references following the requirements of the MDPI publishing house.

Round 2

Reviewer 2 Report

The paper has been significantly improved. I have no further comments. I recommend it for publication in its current form.

Author Response

Thank you for your thorough review, it has made our article better.